# CCP1, a Regulator of Tubulin Post-Translational Modifications, Potentially Plays an Essential Role in Cerebellar Development

**DOI:** 10.3390/ijms24065335

**Published:** 2023-03-10

**Authors:** Bo Pang, Asuka Araki, Li Zhou, Hirohide Takebayashi, Takayuki Harada, Kyuichi Kadota

**Affiliations:** 1Department of Organ Pathology, Faculty of Medicine, Shimane University, 89-1 Enya, Izumo 693-8501, Japan; 2Pathology Division, Shimane University Hospital, Izumo 693-8501, Japan; 3Department of Cellular Neurobiology, Brain Research Institute, Niigata University, Niigata 951-8585, Japan; 4Division of Neurobiology and Anatomy, Graduate School of Medical and Dental Sciences, Niigata University, Niigata 951-8510, Japan

**Keywords:** Purkinje cells (PCs), CCP1/Nna1/*AGTPBP1*, Ataxia and Male Sterility (AMS), polyglutamylation, neurodegenerative

## Abstract

The cytosolic carboxypeptidase (CCP) 1 protein, encoded by *CCP1*, is expressed in cerebellar Purkinje cells (PCs). The dysfunction of CCP1 protein (caused by *CCP1* point mutation) and the deletion of CCP1 protein (caused by *CCP1* gene knockout) all lead to the degeneration of cerebellar PCs, which leads to cerebellar ataxia. Thus, two *CCP1* mutants (i.e., Ataxia and Male Sterility [AMS] mice and Nna1 knockout [KO] mice) are used as disease models. We investigated the cerebellar CCP1 distribution in wild-type (WT), AMS and Nna1 KO mice on postnatal days (P) 7–28 to investigate the differential effects of CCP protein deficiency and disorder on cerebellar development. Immunohistochemical and immunofluorescence studies revealed significant differences in the cerebellar CCP1 expression in WT and mutant mice of P7 and P15, but no significant difference between AMS and Nna1 KO mice. Electron microscopy showed slight abnormality in the nuclear membrane structure of PCs in the AMS and Nna1 KO mice at P15 and significant abnormality with depolymerization and fragmentation of microtubule structure at P21. Using two *CCP1* mutant mice strains, we revealed the morphological changes of PCs at postnatal stages and indicated that CCP1 played an important role in cerebellar development, most likely via polyglutamylation.

## 1. Introduction

The cytosolic carboxypeptidase (CCP) 1, also known as Nna1 (nervous system nuclear protein induced by axotomy protein 1), or *AGTPBP1* (ATP/GTP binding protein 1), mainly encodes an enzyme that deglutamates target proteins, which consist of 789 amino acid residues. The *CCP1* is located on human chromosome 9 at 9q21.33, with a total of 30 exons, and is widely expressed in bone marrow, brain and other 23 tissues. The mouse *CCP1* gene is located at the B2 band or chromosome 13 (chr.13, 31.87 cM), has 31 exons, and is mainly expressed in the cerebellum, cerebral cortex and 21 other tissues (Mouse Genome Informatics (MGI): http://www.informatics.jax.org (accessed on 3 August 2022)). A mutation of *CCP1* leads to the degeneration of cerebellar Purkinje [1] and granule cells [2], which leads to the appearance of a series of pathological signs, such as cerebellar ataxia and cognitive retardation [3].

Several *Agtpbp1* alleles have been reported in mice, including 8 spontaneous gene mutation mouse models (MGI). The AMS (Ataxia and Male Sterility) mouse has an *ams* point mutation [4,5]. It is derived from an autoimmune disease-prone genetic variant strain of MRL/*lpr* mice colony. The AMS mouse is named for its two main clinical manifestations, signs of cerebellar ataxia due to Purkinje cell (PC) degeneration that begin on postnatal day (P) 21 and oligospermia due to defective spermatic differentiation [6]. The maintenance and establishment of the AMS mouse strain can be used to analyze the occurrence and development of the disease in mice before clinical onset.

Cerebellar PCs are among the most prominent neurons in the central nervous system. They have a unique planar fan-shaped dendritic structure with extensive branching [7]. This dendritic morphology is ideal for receiving single (or a few) excitatory synaptic signals from more than 100,000 parallel fibers, another excitatory input to Purkinje neurons that is provided by a climbing fiber that forms hundreds to thousands of synapses with PCs [8]. PCs are the only neurons that send output signals from the cerebellar cortex [5]. The same progressive degeneration of PCs was also demonstrated in Purkinje Cell Degeneration (PCD) mice [9], which have the same allele as AMS mice. Studies have shown that β-tubulin polyglutamate is caused by mutations in the *CCP1* gene [10]. Polyglutamylation is a reversible post-translational modification that sequentially adds glutamic acid residues to its target protein to form side chains [11,12]. Maintaining tubulin homeostasis is the basis for microtubules maintaining the shape and polarity of the nerve cells [13] and participating in cell movement, intracellular material transport and signal transduction [14,15]. Studies have shown that CCP1 can dynamically regulate the balance of polyglutamylation by hydrolyzing the a-carboxyl group of glutamic acid [16,17]. The imbalance of polyglutamylation results in both structural abnormalities and microtubule dysfunction, which eventually lead to neurodegeneration. The Nna1 knockout (KO) mouse used in this report is a PCD mouse model in which exons 21 and 22 on chromosome 13 are artificially knocked out [18]. Sheikh et al. have found an increase in the post-translational modification of polyglutamate in AMS mice [4], which is presumed to underlie PC neurodegeneration. However, the detailed disease progression of neuronal damage caused by CCP1 dysfunction has not yet been elucidated.

In this report, we further monitor the impact of CCP1 dysfunction on the development of different cells in the cerebellum during the pre-onset and clinical stages by examining the wild type (WT), AMS and Nna1 KO mice. This study provides supportive data for pre-onset screening and gene-targeted therapy for cerebellar development-related diseases.

## 2. Results

### 2.1. Ams Mutations and Deletions of CCP1 Alleles Lead to Morphological Changes during the Development of the Mouse Cerebellum

The histopathology of HE-stained cerebellar tissue of WT mice was examined by light microscopy on postnatal days (P) 7, P15, P21 and P28. The immunohistochemical expression of CCP1 in the molecular and the granular layers of the WT mice was the strongest at P7, before structural completion of the Purkinje cell layer, and decreased at P15. The CCP1 expression was relatively stable after P21. Additionally, positivity in the soma and dendrite of the PC was clearly shown at P15 (Appendix A), when the PC layer is discernible, through P28 (Figure 1A). In comparison to WT mice, the PCs in the cerebellum of AMS and Nna1 KO mice started to degenerate from P21, after which the number of PCs decreased and almost completely disappeared by P28 (Figure 1A).

After the statistical analysis of the CCP1 staining intensity in PCs, we found that staining intensity in WT PCs was significantly different from that in AMS and Nna1 KO PCs before P15. However, after the PCs began to degenerate, there was also a statistically significant difference in the intensity of CCP1 staining between the AMS mice and the Nna1 KO mice at P21 and P28 (*p* < 0.01) (Figure 1B).

IF staining in WT mice highlighted the expression of CCP1, first in the granular layer, and subsequently in the soma and dendrite of PCs at P15 after the structural completion of the Purkinje cell layer. The remaining PCs of AMS homozygous mice after the symptomatic stage (P21 and P28) exhibited a CCP1 signal, suggesting the involvement of CCP1 in the postnatal development of the cerebellum. Meanwhile, the CCP1 expression in the granular layer of WT, AMS and Nna1 KO mice was stable from P7 to P28, even though they were in the postsymptomatic stage (Figure 2).

### 2.2. CCP1 Protein Expression during Cerebellar Development in the Two Mutant Mouse Strains

We next explored the expression levels of CCP1 protein in the cerebellum of WT and AMS mice at postnatal stages (P7, P15, P21 and P28). The Western blotting showed that the level of CCP1 protein in the cerebellum of AMS mice was decreased in comparison to WT mice at each of the four different ages. In the Nna1 KO cerebellum, the CCP1 protein was under a detectable level in this condition (Figure 3A).

As the band expression of AMS mice is weak, we next increased the sample size and protein quality to analyze the changes of CCP1 protein levels in AMS mice at different ages. Western blotting showed that the expression level of CCP1 in AMS mice was significantly different from that of the WT from P15 (Figure 3B,C).

### 2.3. Ultrastructural Alterations on PCs Caused by CCP1 Gene Mutation and Knockout

To further confirm the effect of *CCP1* gene alteration on the development of PCs, we used transmission electron microscopy to observe the PCs in the mouse cerebellum at four different ages. Electron microscopy revealed that the dendritic structure of PCs was preserved at P7 in both AMS and Nna1KO mice. However, at P15, the nuclear membranes of PCs in AMS and Nna1 KO mice showed slight shrinkage and a concave morphology. In comparison to WT mice, the Golgi apparatus was slightly swollen, but the mitochondria and microtubules appeared to have normal structures at this age. At P21 in AMS mice, the following features were observed: the electron density of PC dendrites was increased; the mitochondria were swollen, their volume was increased, and their cristae structure was destroyed; the endoplasmic reticulum was expanded to form a vesicle-like structure; the microtubule structure was depolymerized and broken; and some myelin figures and autophagosomes were observed. In addition to the above morphological changes, polysome depolymerization and some lysosomes were also observed in Nna1 KO mouse cerebellar PCs on P21 (Figure 4).

## 3. Discussion

In this report, we show that the loss-of-function of CCP1 caused by the mutation of *CCP1* triggers a series of neurodegenerative changes in the AMS mouse cerebellum. The AMS mouse is currently the only spontaneous autosomal recessive inheritance mouse model in which homozygosity can be easily screened at the pre-onset stage using molecular techniques. This mouse model can be used to reveal the molecular mechanisms of human CCP1 deficiency to analyze the pathological change of different types of neuronal cells, such as PCs and granular cells in cerebellum, as well as extracerebellar cells (e.g., retinal photoreceptor cells) [6], olfactory mitral cells and spermatogenic cells of the testis [5]. We demonstrated that the lack of CCP1 most likely affects the normal structure and function by perturbing the process of posttranslational modification of microtubules, contributing to morphological alteration of the PCs and progressive cerebellar breakdown. Our findings confirmed that the formation of healthy, typical PCs with normal synaptic architecture is essential for maintaining cerebellar motor tasks. 

Munoz-Castaneda et al. reported the use of calbindin-D-28k as a PC marker in PCD mice and noted that the length of PC dendrites was shortened and the soma area was reduced at P22 [19]. Harada et al. found that the number of PCs in AMS mice decreased from P21 [5]. Immunohistochemical analyses of cerebellum specimens from AMS and Nna1 KO mice revealed that the number of PCs decreased from P21; however, no significant differences in CCP1 staining intensity in a single PC were found between WT and AMS mice at P21 and P28 (Figure 1B), indicating that the amount of CCP1 protein is not affected by the vulnerability of PCs.

Previous studies have shown that missense mutations may disrupt the three-dimensional structure of the CCP1 protein by interfering with protein folding and affecting the shape of the active site, resulting in CCP1 protein dysfunction [20]. In our analyses of the CCP1 protein expression in the mouse cerebellum, no expression of CCP1 was found in the cerebellum of Nna1 KO mice; however, the cerebellar CCP1 expression of AMS mice after P15 was lower than that of WT mice. In addition, there was no significant difference in CCP1 levels among the four different ages of AMS mice. The point mutation of AMS mice induced similar symptoms to those observed in the Nna1 KO mice (i.e., Purkinje cell degeneration and ataxia began to appear at P21). These results indicate that the point mutation of AMS mice weakened the CCP1 protein expression, and that the expressed protein also lost its function. Therefore, we speculate that the point mutation of AMS mice leads to the change of arginine to proline at position 808 (CCP1^R808P^), which affects the stability of the tertiary structure of CCP1 protein and eventually leads to loss of function.

The depolymerization and fragmentation of the microtubule structure in the cerebellum of AMS and Nna1 KO mice from P15 was captured by TEM. However, a normal synaptic structure was maintained to P21, indicating that the protein structure of CCP1 is changed by spontaneous *CCP1* mutation or knockout, and that PCs lose the proper CCP1 function leading to the degeneration of PCs at around P15. Probably due to the disturbance of the balance of post-translational modifications of tubulin polyglutamate, microtubule fragmentation causes the organelles in PCs to basically lose their normal structure by P21, which also confirmed our hypothesis. The first signs of cerebellar ataxia in AMS mice also suggest that polyglutamylation was first discovered in brain tubulin [21], where it is highly enriched in comparison to other tissues [22].

The dysregulation of the tubulin–polyglutamylation balance caused by *CCP1* mutations has also been found in human neurodegenerative diseases, where it mainly manifests as progressive neurodegeneration in the central and peripheral nervous systems during infancy. Its most obvious pathological manifestations are cerebellar atrophy and degeneration of the lower motor neurons [3,23,24]. Many of the pathological features described in these patients are recapitulated in AMS and Nna1 KO mice, making these mice excellent models for studying human disease. Furthermore, since AMS mice can be genotyped in the preclinical stage, we focused on observing the predegeneration stage of PCs in mice (i.e., one- and two-week-old mice) to find the mechanism of Purkinje cell degeneration. It was found that although Nna1 KO mice had no protein expression of CCP1 in Western blotting, immunohistochemical and immunofluorescence analyses still recognized the CCP1 signal. This discrepancy may be partly due to the character of the polyclonal CCP1 antibody used in this study; it is possible that the antibody recognizes other homologs of the CCP family in an undenatured form and does not recognize denatured one(s) in Western blotting (Appendix A). CCP1 belongs to the metallocarboxypeptidase M14 family [25,26], and five other *CCP1*-like genes (*CCP2* to *CCP6*) were identified in the mouse genome [27]. These results also partly explain why *CCP1* mutations and gene knockout lead to protein dysfunction, while PCs can still survive for 3 weeks, including cells in other tissues and organs that also progressively degenerate at different stages of mouse development.

We hope that the AMS mouse model can provide pre-onset screening for diseases associated with cerebellar gene defects and provide a longer treatment time window for disease treatment, reducing the impact of disease on irreversible cerebellar damage. It will provide a more stable animal research model for the treatment of disease.

## 4. Materials and Methods

### 4.1. Mice 

The AMS mice described by Harada et al. [5] were bred and maintained in the Institute of Experimental Animals, Shimane University. For the Nna1 KO mice described by Zhou et al. [18], Nna1 heterozygous mice were provided by Prof. Hirohide Takebayashi (Niigata University, Niigata, Japan) and were bred and maintained in the same manner as the AMS mice. The animal care and use for this study were performed in accordance with the recommendations of the Institute of Experimental Animals, Shimane University. Experiential procedures were specifically approved by Animal Experimentation Codes of Shimane University in compliance with the international guidelines (approval numbers: IZ2-32, IZ2-87, IZ2-88). 

### 4.2. Mouse Genotyping 

For genotyping, genomic DNA was isolated from tissue fragments of mouse ear using a High Pure PCR Template Preparation Kit (Roche, Mannheim, Germany). The extracted DNA was used as a template for amplification. Following PCR amplification, the PCR products from AMS mice were digested with a restriction enzyme (NspV) for 1.5 h at 37 °C (this step was skipped in Nna1 KO mice). Then, the PCR products were separated by agarose gel electrophoresis to identify the DNA bands. The forward and reverse primer pairs listed in Table 1 were both used to define AMS mice (Nna1 exon17) and Nna1 KO mice (Nna1-LOX and B2) genotypes.

### 4.3. Tissue Preparation and Morphological Analysis

Mice were deeply anesthetized with pentobarbital (10 mL/kg intraperitoneal injection) and euthanized with a lethal dose of anesthetic agent. The brains were cut along the median sagittal line and immersed in 4% paraformaldehyde in 0.1 M phosphate buffer (pH 7.2) overnight, fixed with 10% formalin for 1 day, then embedded in paraffin until use.

Sections (thickness: 4 µm) from formalin-fixed and paraffin-embedded (FFPE) whole brain tissue obtained from 12 WT mice (*n* = 3 animals in each period) were stained by hematoxylin and eosin (HE) for a histopathological examination. 

Immunohistochemistry was performed using Bond-III stainers (Leica Biosystems, Tokyo, Japan). Sections (thickness: 4 μm) obtained from 36 mice (3 per stage of WT, AMS and Nna1 KO mice) of the same tissue used for the histopathological examination were analyzed according to the following protocol: antigen retrieval 20 min at pH 6, incubation with CCP1 antibody (dilution 1:200, Proteintech, Rosemont, IL, USA) for 15 min, and counterstaining with hematoxylin for 5 min. The semiquantitative analysis of immunohistochemically stained sections was performed with a WinROOF 2015 system [28,29].

For immunofluorescence staining, the 4-μm thick sections of the cerebellum of 27 mice (3 per stage of WT, AMS and Nna1 KO mice) were sequentially treated with xylene (3 times for 5 min) and 95% ethanol (3 times for 5 min). They were then incubated with the primary antibody overnight at 4 °C, washed 3 times in 1× phosphate-buffered saline (PBS), incubated for 1 h in the specific secondary antibody and counterstained with DAPI (1500 ng/mL) to identify the cell nuclei. Finally, the samples were mounted with the Fluoromount/Plus™ (Diagnostic BioSystems, Pleasanton, CA, USA). The samples were then examined with a laser confocal microscope (Olympus CLSM FV1000-D, Tokyo, Japan).

The following antibodies were used: mouse monoclonal antibodies anti-Calbindin-D-28K (dilution 1:5000, Sigma, St. Louis, MO, USA; used to label Purkinje cells (PCs) of the cerebellum [30]) and rabbit polyclonal antibodies anti-CCP1 (dilution 1:100 Proteintech, Rosemont, IL, USA). The secondary antibodies were Donkey anti-mouse IgG H&L conjugated with Alexa Fluor 568 (dilution 1:1000, Abcam, Carlsbad, CA, USA) for anti-Calbindin-D-28K and Goat anti-rabbit IgG H&L conjugated with Alexa Fluor 488 (dilution 1:1000, Abcam, Carlsbad, CA, USA) for anti-CCP1.

### 4.4. Western Blotting

For Western blotting, mouse cerebellum specimens obtained from 36 mice (3 per stage of WT, AMS and Nna1 KO mice) were dissected and lysed in RIPA buffer (Nacalai Tesque, Kyoto, Japan). Homogenates were centrifuged at 10,000 rpm for 10 min at 4 °C, subsequently determined their protein concentration by TaKaRa BCA protein Assay Kit (Takara Bio, Kusatsu, Japan). After mixing with an equal volume of sodium dodecyl sulfate (SDS) sample buffer (0.5M Tris-HCl, pH 6.8, 4% SDS, 20% glycerol, 0.008% bromophenol blue) and denaturing in the presence of 14.24 M 2-mercaptoethanol at 100 °C for 5 min, homogenates containing 4 μg of protein were separated by SDS-polyacrylamide gel electrophoresis (PAGE) and electroblotted onto a polyvinylidene fluoride (PVDF) microporous membrane (Millipore, Carrigtwohill, Ireland). After blocking with 5% skim milk for 1 h, membranes were incubated with the following primary antibodies: rabbit anti-CCP1 (1:1000, Proteintech, Rosemont, IL, USA) and mouse anti-β-actin antibodies (1:5000, Proteintech, Rosemont, IL, USA), followed by HRP-conjugated mouse anti-rabbit IgG (1:1000, Cosmo Bio Co., Tokyo, Japan) and HRP-conjugated sheep anti-mouse IgG antibodies (1:5000, Amersham, Little Chalfont, UK). Signals were obtained using ECL (Bio-Rad, Hercules, CA, USA) and a luminescence image analyzer (ImageQuant 800) [31], and the pixel intensity was analyzed using a luminescence image analyzer and normalized in comparison to the signal intensity of β-actin.

### 4.5. Transmission Electron Microscope

Cerebellum tissue specimens obtained from 27 mice (3 per stage of WT, AMS and Nna1 KO mice) were fixed in 2% glutaraldehyde overnight at 4 °C and rinsed with 0.2 M PB buffer for 20 min. Samples were dehydrated in a graded series of acetone (25%, 50%, 75%, 100%) before being embedded in epoxy resin (TAAB medium grade) and polymerized overnight at 60 °C. Sections were cut on a Leica EM UC7 ultramicrotome. First, semithin sections (0.5 μm) were stained with toluidine blue for light microscopy to identify the area of interest and confirm the orientation of the tissue. Ultrathin sections (70 nm) were then transferred to copper grids, stained with uranyl acetate and lead citrate and examined on a Topcon EM-002B (Tokyo, Japan) transmission electron microscope. Images were captured at ×10,000 magnification.

### 4.6. Statistical Analysis 

All statistical analyses were performed by the SPSS version 26.0 software program with 1-way ANOVA followed by a Tukey post hoc test. Values were given as the mean ± SD. The criterion for statistical significance was set at *p* < 0.05.

## Figures and Tables

**Figure 1 ijms-24-05335-f001:**
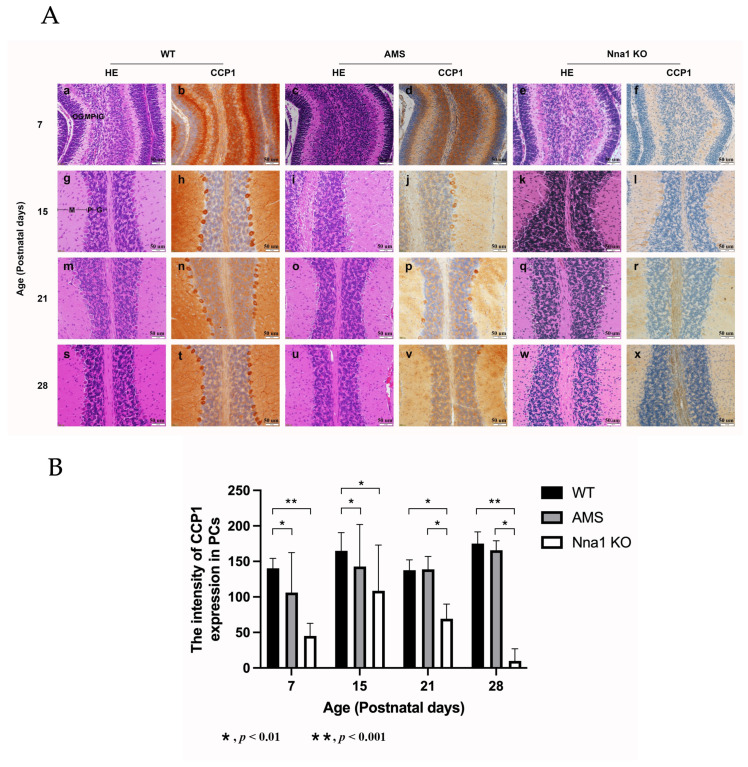
The representative light microscope appearance of the mouse cerebellum on postnatal (P) day P7, P15, P21 and P28. (**A**) HE staining of the wild-type (WT) mouse cerebellum (**a**,**g,m,s**) and corresponding CCP1 immunohistochemistry (IHC) (**b**,**h,n,t**). Age-matched cerebellar CCP1 HE (**c**,**i**,**o**,**u**) and IHC (**d**,**j**,**p**,**v**) of the AMS mice, as well as HE (**e**,**k**,**q**,**w**) and IHC (**f**,**l**,**r**,**x**) of matched Nna1 KO mice, respectively. M, molecular layer; P, Purkinje cell layer; IG, inner granular layer; OG, outer granular layer. Scale bars: 50 µm. (**B**) The intensity of the CCP1 expression in the AMS and Nna1 KO mouse cerebellar Purkinje cells according to age. The data represent the mean value for >3 mice per group ± SD. Bars: black bars—WT; gray bars—AMS; white bars—Nna1 KO mice. * *p* < 0.01, ** *p* < 0.001. CCP1: cytosolic carboxypeptidase. AMS: ataxia and male sterility.

**Figure 2 ijms-24-05335-f002:**
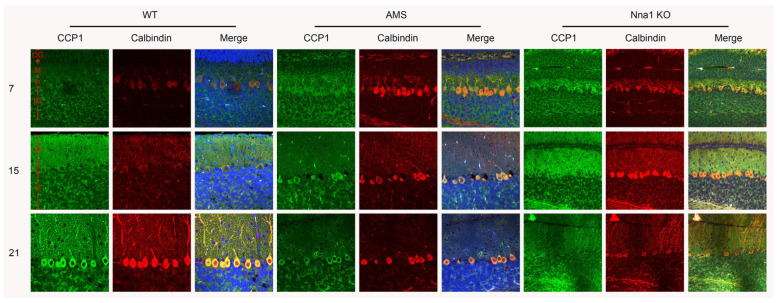
Immunofluorescence detection of CCP1 expression in the Purkinje cells (PCs). Micrograph of PCs in the cerebellum labeled with CCP1 (green) and Calbindin D-28K (red) in the WT (left panels), AMS (middle panels) and Nna1 KO mice (right panels). The number of PCs was reduced in the cerebellum of AMS and Nna1 KO mice in the P21 group, but the expression of CCP1 was still observed in residual PCs. Green, CCP1; Red, Calbindin (Purkinje cell marker); blue, DAPI (counterstaining of nuclei); M, molecular layer; P, Purkinje cell layer; IG, inner granular layer; OG, outer granular layer. CCP1: cytosolic carboxypeptidase. AMS: ataxia and male sterility.

**Figure 3 ijms-24-05335-f003:**
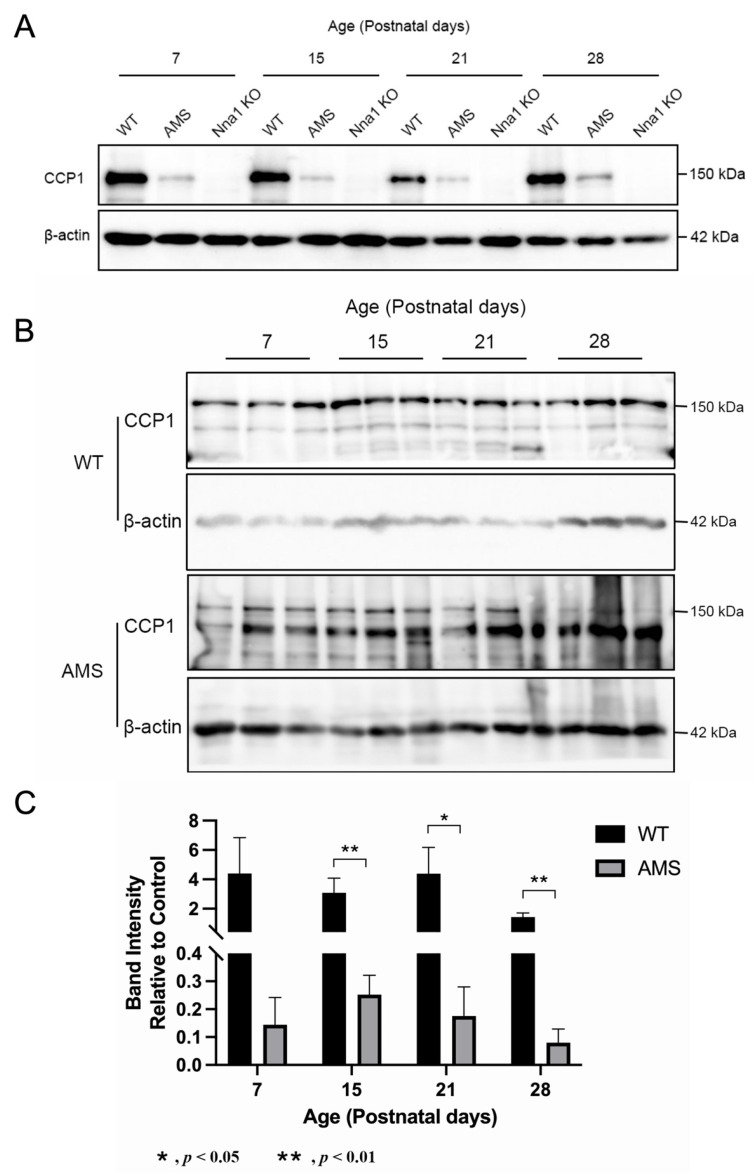
The CCP1 protein expression in the cerebellum of three mouse genotypes. (**A**) The CCP1 protein expression in the cerebellum in four age groups of WT, AMS and Nna1 KO mice. β-actin was used as a loading control. (**B**) The CCP1 expression at 138 kDa in four age groups of WT and AMS mice (*n* = 3 animals in each period). (**C**) The semiquantitative analysis of the CCP1 intensity (mean ± SD) in (**B**). Bars: black bars—WT; gray bars—AMS. * *p* < 0.05, ** *p* < 0.01. CCP1: cytosolic carboxypeptidase. AMS: ataxia and male sterility.

**Figure 4 ijms-24-05335-f004:**
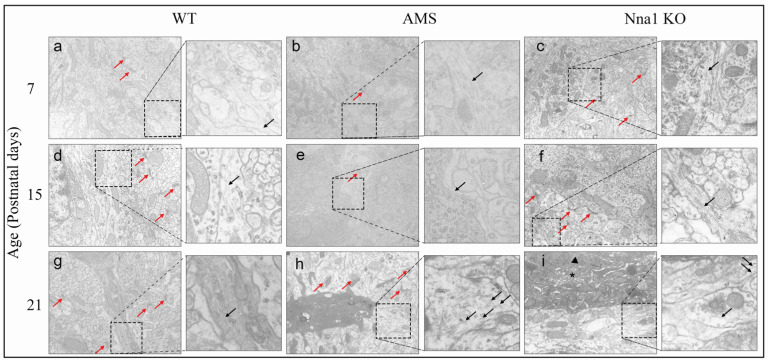
Transmission electron micrographs of microtubule and synapse of cerebellar Purkinje cells in WT (**a**,**d**,**g**), AMS (**b**,**e**,**h**) and Nna1 KO mice (**c**,**f**,**i**). Normal microtubule (black arrows) and synaptic (red arrows) structures in the cerebellum of WT mice at postnatal (P) day P7 (**a**), P15 (**d**), and P21 (**g**). There were no obvious changes of microtubules and synapses in the cerebellum of AMS and Nna1 KO mice at P7 (**b**,**c**) or P15 (**e**,**f**). In PCs of AMS mice, marked synaptic alteration and fragmentation of microtubules were observed at P21 (**h**). In the PCs of Nna1 KO mice, pycnotic nuclei (triangle), swollen Golgi (star) and lysosomal structures were also found at P21 (**i**). Magnification in each image of morphometry (×10,000). AMS: ataxia and male sterility.

**Table 1 ijms-24-05335-t001:** PCR primers used for genotyping.

Name	Forward	Reverse
Nna1 exon17	5′-AATCGGCACAATCCTC-3′	5′-TGAACTGACAGATATGTTCATAG-3′
Nna1-LOX	5′-TTACAGACTCCTGCACCTGG-3′	5′-GAGTCTGACGCATTACCCAC-3′
B2	5′-CTTGTACAAAGTGGCCCGAC-3′	

## Data Availability

All data in this study are shown in the report.

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
