# Peer review of "CCP1, a Regulator of Tubulin Post-Translational Modifications, Potentially Plays an Essential Role in Cerebellar Development"

_ijms, 2023, doi:10.3390/ijms24065335_

Round 1
Reviewer 1 Report
The paper aims to describe the role of CCP1 in cerebellar development using ICC and western blot analysis.
Although this is an interesting area of investigation there are several flaws that make the manuscript unacceptable in this form.
The main issues of concern/missing information are listed below.
What is the ultrastructural distribution of CCP1 protein in wt and mouse models?
As the authors explain positivity to Calbindin in Purkinje cells in AMS and Nna1KO mouse models and not in wt animals at P7-P15?
In Fig.A (b) is quite difficult to detect CCP1 positivity and has different magnification.
CCP1 is expressed only in Purkinje cell soma or also in dendritic tree?
In this report authors describe that CCP1 can dynamically regulate the balance of polyglutamylation and that imbalance of polyglutamylation results in structural abnormalities and microtubule dysfunction.
For this reason, it would have been important to show tubulin distribution in the cerebella of these two animal models.
Author Response
February 28th, 2023
Manuscript ID: ijms-2250445
Type of manuscript: Research article
Title: CCP1, a regulator of tubulin post-translational modifications, potentially plays an essential role in cerebellar development
Dear Editor,
Thank you very much for reviewing our submission titled, “CCP1, a regulator of tubulin post-translational modifications, potentially plays an essential role in cerebellar development (Manuscript ID: ijms-2250445)”. We really appreciate your constructive comments, and revised the manuscript accordingly (marked by track change function in Microsoft word in the revised manuscript). Each comment has been addressed below. We feel these changes have significantly strengthened this manuscript and hope that it will now be suitable for publication in ‘International Journal of Molecular Sciences’.
Comments and answers:
More details of the updated manuscript are required:
Reviewer 1
- What is the ultrastructural distribution of CCP1 protein in wt and mouse models?
Re: We appreciate your meaningful proposal very much.
According to your opinion, we retrieved databases such as NCBI, MGI, and GTEX Portal, but the database has not yet published the specific distribution of CCP1 on the ultrastructural structure. However, according to our understanding, the CCP1 gene is mainly encoded a cytosolic carboxypeptidase 1, which is the main function to shorten the length of the microtubal side chain.
- As the authors explain positivity to Calbindin in Purkinje cells in AMS and Nna1 KO mouse models and not in wt animals at P7-P15?
Re: We are very grateful for your valuable questions.
As we present figure 2, Calbindin D-28K is also expressed in the P7- P15 of WT mice.
The immunohistochemical staining of the Calbindin D-28K in the same age is applied in the text (DOI: 10.1007/s12311-015-0724-2). Therefore, we choose its antibody as a specific positioning of PCs.
- In Fig.A (b) is quite difficult to detect CCP1 positivity and has different magnification.
Re: Thanks a lot for your constructive suggestions.
The method of CCP1 detection intensity and magnification in panel b of Figure 1A are the same as the rest of the panels. Because the cerebellar structure of mice in the P7 was not yet fully developed, the outer granular layer of the cerebellum had not completely migrated to the inner granular layer, and the dendrites of Purkinje cells were shorter. The entire section of the cerebellum was smaller than that of the P15-P28 group, so when the pictures were taken at the same magnification, more lobes of the cerebellum were collected in the pictures of the mice in the P7.
- CCP1 is expressed only in Purkinje cell soma or also in dendritic tree?
Re: We appreciate your important advice deeply.
CCP1 is expressed in the both of Purkinje cell soma and dendritic. We could add a high-magnification Purkinje cell figure in the supplementary materials section.
- In this report authors describe that CCP1 can dynamically regulate the balance of polyglutamylation and that imbalance of polyglutamylation results in structural abnormalities and microtubule dysfunction. For this reason, it would have been
February 28th, 2023
Manuscript ID: ijms-2250445
Type of manuscript: Research article
Title: CCP1, a regulator of tubulin post-translational modifications, potentially plays an essential role in cerebellar development
Dear Editor,
Thank you very much for reviewing our submission titled, “CCP1, a regulator of tubulin post-translational modifications, potentially plays an essential role in cerebellar development (Manuscript ID: ijms-2250445)”. We really appreciate your constructive comments, and revised the manuscript accordingly (marked by track change function in Microsoft word in the revised manuscript). Each comment has been addressed below. We feel these changes have significantly strengthened this manuscript and hope that it will now be suitable for publication in ‘International Journal of Molecular Sciences’.
Comments and answers:
More details of the updated manuscript are required:
Reviewer 1
- What is the ultrastructural distribution of CCP1 protein in wt and mouse models?
Re: We appreciate your meaningful proposal very much.
According to your opinion, we retrieved databases such as NCBI, MGI, and GTEX Portal, but the database has not yet published the specific distribution of CCP1 on the ultrastructural structure. However, according to our understanding, the CCP1 gene is mainly encoded a cytosolic carboxypeptidase 1, which is the main function to shorten the length of the microtubal side chain.
- As the authors explain positivity to Calbindin in Purkinje cells in AMS and Nna1 KO mouse models and not in wt animals at P7-P15?
Re: We are very grateful for your valuable questions.
As we present figure 2, Calbindin D-28K is also expressed in the P7- P15 of WT mice.
The immunohistochemical staining of the Calbindin D-28K in the same age is applied in the text (DOI: 10.1007/s12311-015-0724-2). Therefore, we choose its antibody as a specific positioning of PCs.
- In Fig.A (b) is quite difficult to detect CCP1 positivity and has different magnification.
Re: Thanks a lot for your constructive suggestions.
The method of CCP1 detection intensity and magnification in panel b of Figure 1A are the same as the rest of the panels. Because the cerebellar structure of mice in the P7 was not yet fully developed, the outer granular layer of the cerebellum had not completely migrated to the inner granular layer, and the dendrites of Purkinje cells were shorter. The entire section of the cerebellum was smaller than that of the P15-P28 group, so when the pictures were taken at the same magnification, more lobes of the cerebellum were collected in the pictures of the mice in the P7.
- CCP1 is expressed only in Purkinje cell soma or also in dendritic tree?
Re: We appreciate your important advice deeply.
CCP1 is expressed in the both of Purkinje cell soma and dendritic. We could add a high-magnification Purkinje cell figure in the supplementary materials section.
- In this report authors describe that CCP1 can dynamically regulate the balance of polyglutamylation and that imbalance of polyglutamylation results in structural abnormalities and microtubule dysfunction. For this reason, it would have been important to show tubulin distribution in the cerebella of these two animal models.
Re: Thank you so much for your crucial comment.
According to your opinion, we will partially amplify the microtubility part of Figure 4 to facilitate readers' more intuitive observation results.
important to show tubulin distribution in the cerebella of these two animal models.
Re: Thank you so much for your crucial comment.
According to your opinion, we will partially amplify the microtubility part of Figure 4 to facilitate readers' more intuitive observation results.

Reviewer 2 Report
Pang et al., in this manuscript, have examined the role of CCP1, a deglutamylase, in cerebellar and Purkinje cell (PC) functions. Mutations in the gene encoding for CCP1 lead to degeneration of PCs and cerebellar ataxia. The authors use two mouse models; Ataxia and Male Sterility (AMS) model that has a spontaneous point mutation in the gene encoding for CCP1 and CCP1 KO (also known as Nna1 KO) model that lacks the protein in its entirety. The authors demonstrate a loss of PCs across postnatal day 28 (p28) in these mouse models compared to WT controls. The authors attribute this loss to microtubule dysfunction mediated by alterations in post-translational modifications of tubulin, namely polyglutamylation.
While the manuscript aims to address the role of CCP1 in cerebellar and PC development, additional work is required to make a conclusive statement (see specific comments below).
Major comment:
1. While the authors try to address the microtubule deglutamylation role of CCP1 in mediating the loss of PCs in AMS and Nna1 KO mice, the evidence they provide is weak and insufficient. The authors’ main evidence for altered microtubules are the images from the electron microscopy. These images are not clear and it is hard to see visually the microtubule depolymerization and fragments that they are addressing in Fig. 4. Please provide more clear and magnified images of the TEMs. Furthermore, if polyglutamylation of microtubules is indeed affected in the AMS and/or CCP1 mice, this can be addressed directly by IF using polyglutamylated tubulin antibody in the PCs. This can also be addressed by performing WB of the cerebellum of these mice using the same antibody. While major differences may not be observed in the AMS model but minor differences can be picked up and in the Nna1 KO mice the difference should be clear if the authors’ theory is correct. Without this direct proof, the authors cannot assign the deglutamylation role of CCP1 to be responsible for PC death during development.
2. Figure 1 and figure 2 seem to be contradicting one another. In figure 1B, the authors clearly show a reduction in CCP1 intensity in PCs. However, in figure 2, where PCs are labeled with calbindin, despite there being a reduction in CCP1 in the AMS model, there is no reduction in CCP1 in the CCP1 (or Nna1) KO mice. How do the authors explain this? The authors try to address this in the discussion by stating that even though the gene has been knocked out, it is possible that smaller protein fragments may exist. If that is the case, one should observe smaller molecular weight bands in WB (for example in figure 3A) which we don’t. Furthermore, if there are indeed smaller molecular weight fragments, is that what is being picked up in figure 2? However why don’t we observe it in figure 1? Please make the discrepancy between figure 1 and 2 clear.
3. In figure 3, the authors have labeled CCP1 in WT and AMS mice by WB. The WB was performed on two separate membranes. One cannot compare band intensities from different membranes even if you normalize to the control gene. I understand the limitation of wells to run all samples on one gel (membrane). Since the authors are comparing within ages, they should run the WT and AMS p7 on the same gel to compare results (similar for other ages). While this will not affect the results dramatically, this is the technically correct way of doing WB.
Minor concerns:
1. In figure 1, please show H&E staining for AMS and Nna1 KO mice
2. In the discussion, the authors state (in line 172) “Moreover, this degeneration of PCs led not only to motor defects but also to gradual cognitive impairment, which was directly related to the progression of cellular damage”. Since the authors do not demonstrate any motor or cognitive impairment this sentence is misleading. The following sentence of “implied involvement of the cerebellum…” is more appropriate. However line 172-173 makes it seem like the authors demonstrated this experimentally (which they did not).
Author Response
February 28th, 2023
Manuscript ID: ijms-2250445
Type of manuscript: Research article
Title: CCP1, a regulator of tubulin post-translational modifications, potentially plays an essential role in cerebellar development
Dear Editor,
Thank you very much for reviewing our submission titled, “CCP1, a regulator of tubulin post-translational modifications, potentially plays an essential role in cerebellar development (Manuscript ID: ijms-2250445)”. We really appreciate your constructive comments, and revised the manuscript accordingly (marked by track change function in Microsoft word in the revised manuscript). Each comment has been addressed below. We feel these changes have significantly strengthened this manuscript and hope that it will now be suitable for publication in ‘International Journal of Molecular Sciences’.
Comments and answers:
More details of the updated manuscript are required:
Reviewer 2
- While the authors try to address the microtubule deglutamylation role of CCP1 in mediating the loss of PCs in AMS and Nna1 KO mice, the evidence they provide is weak and insufficient. The authors’ main evidence for altered microtubules are the images from the electron microscopy. These images are not clear and it is hard to see visually the microtubule depolymerization and fragments that they are addressing in Fig. 4. Please provide more clear and magnified images of the TEMs. Furthermore, if polyglutamylation of microtubules is indeed affected in the AMS and/or CCP1 mice, this can be addressed directly by IF using polyglutamylated tubulin antibody in the PCs. This can also be addressed by performing WB of the cerebellum of these mice using the same antibody. While major differences may not be observed in the AMS model but minor differences can be picked up and in the Nna1 KO mice the difference should be clear if the authors’ theory is correct. Without this direct proof, the authors cannot assign the deglutamylation role of CCP1 to be responsible for PC death during development.
Re: Thank you so much for your crucial comment.
According to your valuable comments, we partially enlarge the microtubule structure in Figure 4. Because we also want to show the changes of synapses in the process of Purkinje cell degeneration, the original picture is kept. And because the electron microscope picture is too Large, we also consider grouping pictures into supplementary data for readers to observe. At the same time, we are also very grateful for your suggestions. We also considered the detection of polyglutamic by WB, but because other research groups have published results in this area (related WB can refer to literature DOI:10.1111/jnc.14591), we did not repeat this part of the experiment.
- Figure 1 and figure 2 seem to be contradicting one another. In figure 1B, the authors clearly show a reduction in CCP1 intensity in PCs. However, in figure 2, where PCs are labeled with calbindin, despite there being a reduction in CCP1 in the AMS model, there is no reduction in CCP1 in the CCP1 (or Nna1) KO mice. How do the authors explain this? The authors try to address this in the discussion by stating that even though the gene has been knocked out, it is possible that smaller protein fragments may exist. If that is the case, one should observe smaller molecular weight bands in WB (for example in figure 3A) which we don’t. Furthermore, if there are indeed smaller molecular weight fragments, is that what is being picked up in figure 2? However why don’t we observe it in figure 1? Please make the discrepancy between figure 1 and 2 clear.
Re: We appreciate your important suggestions.
CCP1 has a lot of isoforms, including CCP2-CCP6, and the size of its encoded protein varies. Since the Figure 3A and 3B are Intercept the area near our purpose protein, the remaining bands are not observed. Complete WB film results are attached to the supplementary material.
In Figure 1, there is still a small amount of CCP1 expression in the cerebellum of Nna1 KO mice, which is consistent with Figure 2. In theory, knocking out the CCP1 gene will lead to loss of expression of CCP1 protein (as shown in the Nna1 KO group in Figure 3A ). So we consider what we picked up in Figure 1 and Figure 2 as other isoforms of CCP1.
- In figure 3, the authors have labeled CCP1 in WT and AMS mice by WB. The WB was performed on two separate membranes. One cannot compare band intensities from different membranes even if you normalize to the control gene. I understand the limitation of wells to run all samples on one gel (membrane). Since the authors are comparing within ages, they should run the WT and AMS p7 on the same gel to compare results (similar for other ages). While this will not affect the results dramatically, this is the technically correct way of doing WB.
Re: Thank you very much for your valuable comments.
Your suggestion is very valuable to our future research work, but our experimental samples have been preserved for more than two years this time, so it is difficult to conduct repeated experiments in groups. I hope to obtain your understanding.
- In figure 1, please show H&E staining for AMS and Nna1 KO mice
Re: Thanks a lot for the rigorous comments.
We have added HE staining pictures of AMS and Nna1KO mice based on your suggestion.
- In the discussion, the authors state (in line 172) “Moreover, this degeneration of PCs led not only to motor defects but also to gradual cognitive impairment, which was directly related to the progression of cellular damage”. Since the authors do not demonstrate any motor or cognitive impairment this sentence is misleading. The following sentence of “implied involvement of the cerebellum…” is more appropriate. However line 172-173 makes it seem like the authors demonstrated this experimentally (which they did not).
Re: We appreciate your important advice deeply.
Based on your comments, we have revised the sentence and added references (page 7, line 175-180).

Round 2
Reviewer 1 Report
The bibliography can be improved.
Author Response
March 7th, 2023
Manuscript ID: ijms-2250445
Type of manuscript: Research article
Title: CCP1, a regulator of tubulin post-translational modifications, potentially plays an essential role in cerebellar development
Dear Editor,
Thank you very much for reviewing our submission titled, “CCP1, a regulator of tubulin post-translational modifications, potentially plays an essential role in cerebellar development (Manuscript ID: ijms-2250445)”. We really appreciate your constructive comments, and revised the manuscript accordingly (marked by track change function in Microsoft word in the revised manuscript). Each comment has been addressed below. We feel these changes have significantly strengthened this manuscript and hope that it will now be suitable for publication in ‘International Journal of Molecular Sciences’.
Comments and answers:
More details of the updated manuscript are required:
Reviewer 1
- The bibliography can be improved.
Re: We appreciate your important advice.
According to your suggestion, we have added some references and revised the format of the bibliography according to the requirements of the journal.

Reviewer 2 Report
Regarding point number 1: Since the authors aren't showing glutamylation effects directly, I would suggest to word their conclusions more cautiously. Instead of directly attributing polyglutamylation, the authors should state "most likely due to polyglutamylation"
Regarding point number 2: Please include the above explanation of CCP1 multiple isoforms in the manuscript (perhaps in the discussion). This will make it easier for the readers to understand the results
Author Response
March 7th, 2023
Manuscript ID: ijms-2250445
Type of manuscript: Research article
Title: CCP1, a regulator of tubulin post-translational modifications, potentially plays an essential role in cerebellar development
Dear Editor,
Thank you very much for reviewing our submission titled, “CCP1, a regulator of tubulin post-translational modifications, potentially plays an essential role in cerebellar development (Manuscript ID: ijms-2250445)”. We really appreciate your constructive comments, and revised the manuscript accordingly (marked by track change function in Microsoft word in the revised manuscript). Each comment has been addressed below. We feel these changes have significantly strengthened this manuscript and hope that it will now be suitable for publication in ‘International Journal of Molecular Sciences’.
Comments and answers:
More details of the updated manuscript are required:
Reviewer 2
- Since the authors aren't showing glutamylation effects directly, I would suggest to word their conclusions more cautiously. Instead of directly attributing polyglutamylation, the authors should state "most likely due to polyglutamylation."
Re: Thanks a lot for your constructive suggestions.
According to your suggestion, we have modified some sentences in the conclusion part to make the article more rigorous (page 7, line 173; page 7, line 203).
- Please include the above explanation of CCP1 multiple isoforms in the manuscript (perhaps in the discussion). This will make it easier for the readers to understand the results.
Re: We are very grateful for your valuable questions.
We have revised the discussion section to add information about other isoforms of CCP1 (page 8, line 220-225). In order to facilitate readers to obtain clearer information from this article.
